# Materials for High Temperature Liquid Lead Storage for Concentrated Solar Power (CSP) Air Tower Systems

**DOI:** 10.3390/ma14123261

**Published:** 2021-06-12

**Authors:** Antonio Rinaldi, Giuseppe Barbieri, Eduard Kosykh, Peter Szakalos, Claudio Testani

**Affiliations:** 1Laboratory SSPT-PROMAS-MATPRO, ENEA CR Casaccia, Via Anguillarese, 301, S.M. di Galeria, I-00123 Rome, Italy; antonio.rinaldi@enea.it (A.R.); giuseppe.barbieri@enea.it (G.B.); 2Walter Tosto S.p.A. (WTO), Via Erasmo Piaggio, 62, I-66100 Chieti, Italy; e.kosykh@waltertosto.it; 3Department Materials Science and Engineering, Royal Institute of Technology (KTH), Brinellvägen, SE-100 44 Stockholm, Sweden; p.szakalos@kth.se; 4Consorzio CALEF, ENEA CR Casaccia, Via Anguillarese 301, S.M. di Galeria, I-00123 Rome, Italy

**Keywords:** solar concentrated towers, FeCrAl alloys, solar system, alloy 800H, lead corrosion

## Abstract

Today the technical limit for solar towers is represented by the temperature that can be reached with current accumulation and exchange fluids (molten salts are generally adopted and the max temperatures are generally below 600 °C), even if other solutions have been suggested that reach 800 °C. An innovative solution based on liquid lead has been proposed in an ongoing experimental project named Nextower. The Nextower project aims to improve current technologies of the solar sector by transferring experience, originally consolidated in the field of nuclear plants, to accumulate heat at higher temperatures (T = 850–900 °C) through the use of liquid lead heat exchangers. The adoption of molten lead as a heat exchange fluid poses important criticalities of both corrosion and creep resistance, due to the temperatures and structural stresses reached during service. Liquid lead corrosion issues and solutions in addition to creep-resistant material selection are discussed. The experimental activities focused on technical solutions adopted to overcome these problems in terms of the selected materials and technologies. Corrosion laboratory tests have been designed in order to verify if structural 800H steel coated with 6 mm of FeCrAl alloy layers are able to resist the liquid lead attack up to 900 °C and for 1000 h or more. The metallographic results were obtained by mean of scanning electron microscopy with an energy dispersive microprobe confirm that the 800H steel shows no sign of corrosion after the completion of the tests.

## 1. Introduction

This paper outlines the results obtained under the H2020 research project entitled: “Advanced materials solutions for next generation high efficiency concentrated solar power (CPS) tower systems”, of which the acronym is NEXTOWER. The pilot plant is design for a maximum power of 100 kW.

Today, one of the technical limits for the efficiency of most common solar towers is the temperature (T_max_ = 565 °C), which can be reached with current storage and exchange fluids (based on molten salts solution) [1,2,3] even if there are other interesting solutions that have been suggested in the literature for service up to 800 °C [4].

Research activities have focused on providing new solutions for improving current technologies for the production of electricity by concentrating the solar energy, finding a transversal application on experience that has already been consolidated in the field of liquid lead heat exchangers for nuclear use.

The actual innovation consists in the achievement of a higher service temperature (T > 800 °C) in comparison with that of the liquid lead nuclear plants, where the maximum use temperatures are under T = 600 °C, even if further developments are expected to achieve up to 650 °C [5]. These lower temperature ranges represent a mitigation effect of corrosion and creep issues.

Activities are carried out, focused on the importance of achieving a more efficient high-temperature receiver and more efficient heat-storage devices to increase the best conversion of solar to thermal energy [6]. Following this path, the present research is focused on the development of a highly efficient thermal energy management by mean of a liquid lead solution. From a structural point of view, steel selection, however, is also another critical issue to be preliminary solved before starting the realization of an innovative plant, and has already been discussed and shown by several researchers [7,8,9], because of the high temperatures reached. The service temperature in the proposed configuration can easily reach temperatures higher than 800 °C and this is the reason for the use of liquid lead as the primary medium for heat-storing and heat-exchange fluid.

In synthesis, the adoption of liquid lead as a storage material and heat exchange fluid is interesting from a technological point of view, but poses at least two main problems that need to be addressed and solved:Poor corrosion resistance of most commercial Ni-rich high strength steels to chemical attack by lead (nickel is dissolved in lead) [10,11];Resistance to creep deformation, which is activated by the inner service pressure (10 bar) and the high operating temperatures (T > 800 °C) that can be reached, combined with the stresses due to the high specific weight of the fluid itself [12,13].

As mentioned, molten lead attacks steels with a corrosion phenomenon that preferentially attacks the nickel sites contained in the solid solution in structural steels [14].

Moreover, the faced problems were complex because few works in the literature report liquid lead corrosion studies (following Pramutadi [15]) at temperatures of 600 °C and around 700 °C [16,17,18,19,20,21,22,23,24], and, in any case, no data regarding corrosion and adopted solutions were found in the literature for temperatures higher than 715 °C (the case of Short and coauthors [25]).

It must be noted that there is a vast literature with many laboratory test results, but all these studies analyzed test conditions for steels and stainless steels at temperatures lower than those reported here [23,24,25,26,27].

The importance of the subject is represented by issues that have already been reported, such as by Kondo [28], where a detailed metallurgical analysis was reported for a corroded tube under lead–bismuth corrosion attack.

The literature has already put forth evidence of the fact that Al-enriched alloys [29,30,31], and in some cases ceramic/alumina-based solution [32], can be a solution to reduce the corrosion in a stagnant liquid lead–bismuth bath. The same route was also studied by Weisenburger et al. [29], where a positive Al effect addition, as a corrosion barrier for structural steels, was shown. Takaya et al. [30] discussed the corrosion resistance of high duty oxide-dispersed strengthened steel with a high Cr content in stagnant lead–bismuth.

Several solutions for the oxygen control effect have been proposed following Li [26] and taken into consideration for engineering solutions adopted for a prototype of an innovative solar tower and heat exchanger system, named SOLEAD. Moreover, Al-enriched steels (FeCrAl-based) that are able to protect themselves by forming a layer of aluminum oxide were the preferred candidates [32]; this was better described and discussed by Rivai [33] and Ejenstam et al. [34].

The scheme in Figure 1 shows that the solar rays are collected from the field by a receiver that heats up air at a pressure not exceeding 2 bar. The closed-loop air is the primary fluid to heat up the liquid lead in the SOLEAD vessel. A secondary heat exchanger is used to force cold air into the SOLEAD vessel (that contains liquid lead as the heat-storage system) to extract heat from the liquid lead, again heating up the air (this time at a pressure of about 10 bar) to be used by the energy generation unit (a small air/gas turbine with a dynamo) which is not part of the activities.

The experimental activities carried out show the technical solutions adopted to overcome creep and corrosion issues. Corrosion laboratory tests have been designed in order to verify the corrosion resistance of 800H steel up to 900 °C and for 1000 h after coating with 6 mm of FeCrAl. In order to verify if the coating process could activate a deep-element diffusion at the 800H steel interface due to the dilution zone, scanning electron microscopy with an energy dispersive microprobe was used for element profile concentration measurements. The results confirmed a dilution (Ni content under 1.5 wt% after 1 mm) and this solution was identified as a solution for the pilot plant working with liquid lead at T > 800 °C.

## 2. Materials and Methods

The experimental activities, based on previous works at lower temperatures [29,30,31], is focused on the demonstration of FeCrAl’s resistance to liquid lead corrosion attack, even at T = 850 °C.

The solution designed for building the core (named SOLEAD) of an advanced and efficient concentrated solar power (CPS) tower pilot plant, based on liquid lead as a storage and heat exchange fluid, consisted in the selection of one structural steel (commercial 800H, ATI Specialty Rolled Products, New Bedford, MA, US) that could be the structural base for creep-resistance, to have all the surfaces exposed to liquid lead coated with a FeCrAl alloy [32,33]. The coating layer was developed starting from an existing class of FeCrAl family and measurements of the dilution zone along with Ni-diffusion (preferential corrosion site) [32,33] at the interface of the 800H structural steel was assessed using a scanning electronic microscope (SEM–FEG LEO 1550 ZEISS (McQuairie, London, UK)) equipped with an EDS OXFORD X ACT system (v2.2, 2001, Oxford Instruments, Abingdon, UK).

The MatCalc^®^ simulation, (Version 6.02, 2018, MatCalc Engineering, Wien, Austria) for the chemical composition of 800H steel gave an indication of the reinforcing phase active in the operative temperature range (T > 800 °C).

## 3. Results and Discussion

As mentioned, molten lead tends to attack and corrode even stainless steels, especially those enriched with nickel. In fact, Ni-sites are the preferential corrosion starting points [32,33].

A comparison between the effects of exposure at 550 °C for 10,000 h for a sample of AISI 316L and a FeCrAl alloy (Kanthal APMT, Hallstahammar, Sweden) is shown in Figure 2. The AISI 316L sample shows clear corrosion attack for a depth of about 0.3 mm (Figure 2a), while there is no evidence of corrosion for the Kanthal APMT FeCrAl steel sample (Figure 2b).

This fact indicated that, in order to use molten lead at temperatures not lower than T = 850 °C, it is necessary to protect the surface of a structural steel with alumina-former alloys.

In an actual case, again, the literature indicates FeCrAl liquid-lead resistance to be due to the development of two steels based on the FeCrAl system, such as modification of the Khantal family’s chemical composition.

Table 1 reports the nominal chemical compositions of the two FeCrAl-based developed alloys, named Nextower 1 and Nextower 2 (Kanthal APMT, Hallstahammar, Sweden).

Another issue to be solved is temperature measurement. In fact, the thermocouples (G-Sensor S.r.l., Orbassano, Italy) are submerged in liquid lead, so special thermocouple external tubing has been manufactured using commercial Kanthal F alloy (Figure 3). The thermocouple tubing (external diameter of 5.6 mm, thickness 0.8 mm) was tested in liquid lead at T = 850 °C for about 1000 h with excellent results.

The as-fabricated Khantal F tube had a rough surface due to the hot-extrusion manufacturing process (Figure 3a) and it was necessary to grind one side of the surface in order to have a better and smooth surface for any corrosion-evidence study (Figure 3a and Figure 4).

Structural material selection for the main vessel (named SOLEAD S200) was another important technological issue. In fact, the vertical vessel (see SOLEAD details in Figure 1) contained, in its inner volume, two heat exchangers: primary (hot air from the solar receivers versus liquid lead) and secondary (liquid lead versus hot air to the turbine).

The operative conditions were designed to: T = 850 °C and P = 10 bars, which corresponded to a max vessel wall-loop stress of 15 MPa. The weight of the SOLEAD vessels reached about 42 tons, with about 39 tons being liquid lead (at T = 850 °C).

All the commercial and common structural steels that were taken into consideration have a high nickel content, as shown in Table 2, which reports a list of the nominal chemical compositions of the preliminary selection of alloys.

The most critical material property taken into account for the structural design was the creep resistance. Moreover, other economic factors, such as cost, weldability, and market availability, were considered for the final selection.

In fact, any of the FeCrAl low-nickel stainless steels can sustain high-temperature creep stresses for 100,000 h of service. In Table 3 some of the most interesting creep properties are reported for comparisons between the materials. Several contributions, such as manufacturability, commercial availability, and costs, and even the technological knowledge of the research group in terms of manufacturing/welding and operations were taken into account; the final best choice was 800H steel.

In fact, the operative conditions were designed to be T = 850 °C and P = 10 bar, which correspond to a max vessel wall-loop stress of 15 MPa and the 800H steel creep stress at T = 815 °C and 100,000 h for this material was σ = 27 MPa (Table 3). The ASME VIII code was adopted for the structural design and confirmed that 800H steel can be considered suitable for use in service.

In order to have an indication of the metallurgical resistance root of 800H the MatCalc^®^ code was used for a better understanding the second phases present at equilibrium for the desired temperatures.

The MatCalc^®^ simulation confirmed that the γ′ reinforcing phase of 800H alloy was quite stable until 825 °C; above this temperature threshold, it started to dissolve (Figure 5) and this can justify the reason for the literature and standard data using up to 815 °C [39].

Even if the mechanical resistance issue has been faced and solved by adopting 800H steel, the high nickel content of 800H steel still requires coating with FeCrAl alloy on all surfaces facing liquid lead.

After coating, the Ni element diffusion from the 800H steel to the external FeCrAl Nextower alloy coatings was measured.

Two 800H steel laboratory samples were coated with Nextower1 or Nextower2 FeCrAl alloy by mean of gas tungsten overlay-welding techniques at Walter Tosto S.p.A. plants in Chieti. Samples A and B, had the following dimension: 500 mm × 300 mm and 20 mm in thickness. The two samples were coated, respectively, with about 5 mm of Nextower1 or Nextower2 FeCrAl alloy.

Figure 6 shows sample A coated with Nextower1 FeCrAl alloy. The deposition parameters adopted for all the tests are reported in Table 4.

The FeCrAl alloys were prepared by Kanthal in two industrial-scale batches of 10-ton furnaces at the Hallstahammar steelworks in Sweden.

Samples, machined from both the coated samples (A and B), before exposure to liquid lead, were prepared for metallographic examination using a scanning electronic microscope (SEM–FEG LEO 1550 ZEISS (McQuairie, London, UK)) equipped with an EDS OXFORD X ACT system (v2.2, Abington, UK). The element diffusion profiles across the optical interface, between the bulk and coating materials, is reported for sample A in Figure 7. Similar results were recorded for sample B, indicating that the diffusion kinetics are similar for the two Nextower 1 and 2 FeCrAl alloys coated on the 800H steel.

Nextower1 was preferred because no cracking indications appeared after deposition, while an very small indication was detected on Sample B. As can be realized from the profiles in Figure 7, the Ni diffusion content at about 500 μm was under 4% weight and around 1.5 weight % at 1 mm from the inner steel–FeCrAl interface (Figure 6b) and about 4 mm from the external surface.

This last value is regarded and expected as a good nickel content value to avoid heavy corrosion attack from liquid lead [36]. This result, coupled with the corrosion tests carried on Nextower 1 FeCrAl alloy, represent an innovative solution for the manufacturing of liquid-lead-facing components at a temperature of T = 850 °C for 1000 h, which has not yet been discussed in the literature.

## 4. Conclusions

The results in this paper show the main technical problems solved during the selection of materials for manufacturing of a high-temperature liquid-lead (T about 800 °C) heat exchanger and storage unit, named SOLEAD.

The 800H steel was selected for mechanical creep stress resistance.

The laboratory results showed that the FeCrAl alloy, named Nextower 1, was suited for service-facing liquid-lead corrosion resistance at temperatures higher than 800 °C for 1000 h and this represents a solution never obtained or suggested previously in the literature.

The FeCrAl-based alloys, both in bulk and in coated layer forms, were successufully adopted. The experimental full-scale prototype SOLEAD is at the final manufacturing stage at the: Italian Agenzia Nazionale per le nuove tecnologie, l’energia e lo sviluppo economico sostenibile (ENEA) research center in Brasimone, Bologna, Italy.

## Figures and Tables

**Figure 1 materials-14-03261-f001:**
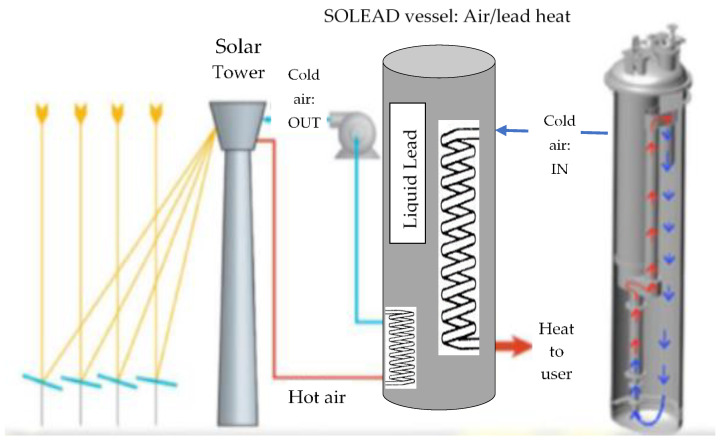
NEXTOWER system with the air/liquid lead (SOLEAD) heat exchanger, detailed on the right.

**Figure 2 materials-14-03261-f002:**
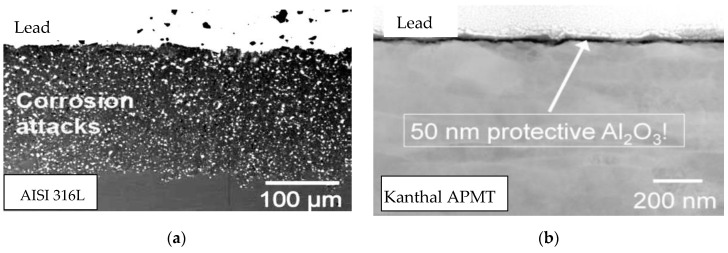
Liquid lead corrosion at T = 850 °C. Comparison of AISI 316L steel without (**a**) and with (**b**) an alumina-former steel.

**Figure 3 materials-14-03261-f003:**
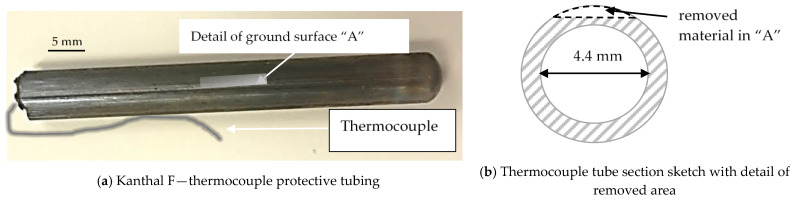
Kanthal F thermocouple tubing ground on a side after exposure to liquid lead for 1000 h at T = 850 °C.

**Figure 4 materials-14-03261-f004:**
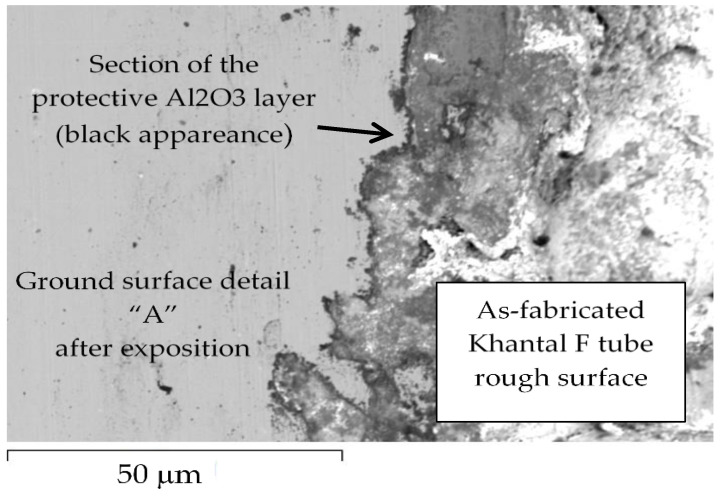
SEM images of the Kanthal F tube exposed to 850 °C stagnant liquid lead. No corrosion attacks were observed across the analyzed ground surface zone, “detail A”. The arrows point to a dark line at the metal–oxide interface, indicating formation of a protective alumina scale. The rough surface is not due to corrosion, but is due to the as-fabricated hot extrusion process.

**Figure 5 materials-14-03261-f005:**
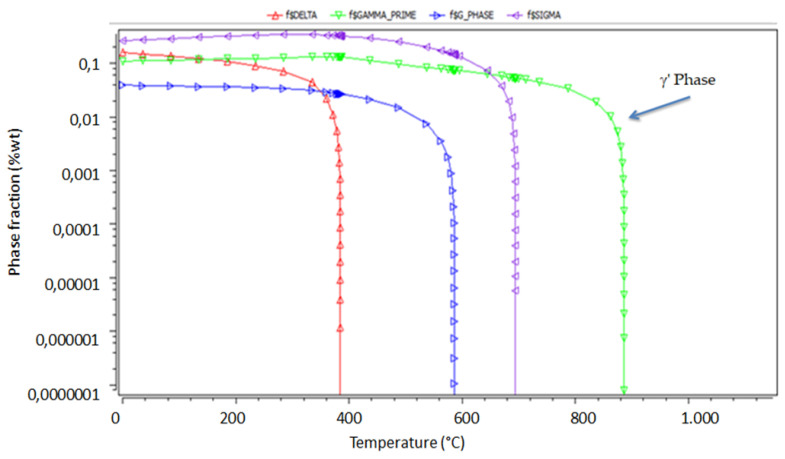
MatCalc^®^ simulation results for 800H steel. Precipitate equilibrium dissolution temperatures.

**Figure 6 materials-14-03261-f006:**
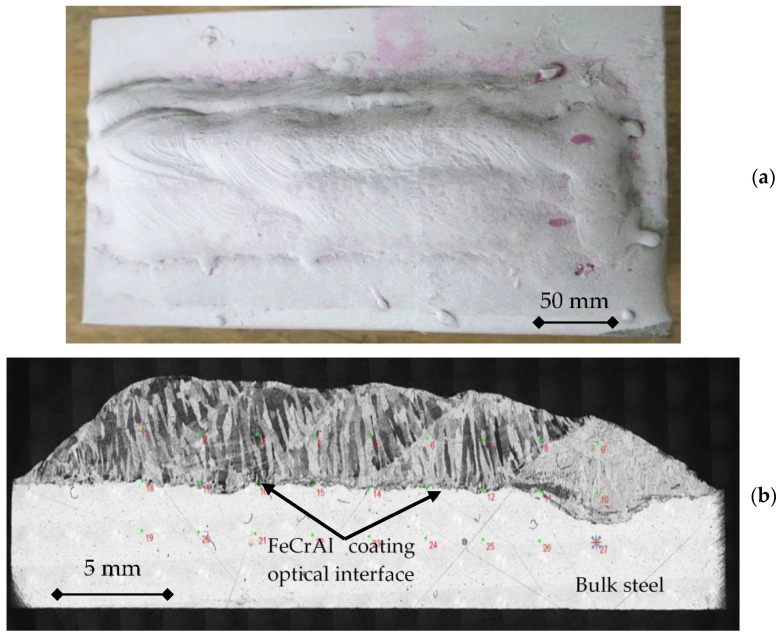
The 800H steel coating (**a**) and a metallographic section of bulk and coating (**b**).

**Figure 7 materials-14-03261-f007:**
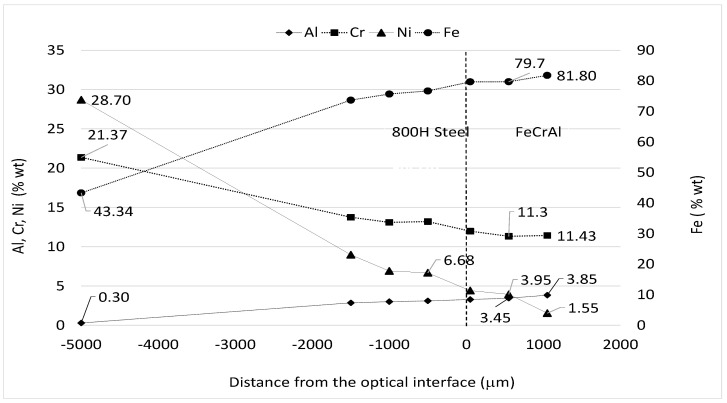
EDS chemical element diffusion across the optical interface (dashed line at point 0) of 800H steel coated with Kanthal Nextower 1.

**Table 1 materials-14-03261-t001:** Nominal chemical composition of the two alloys, Nextower 1 and Nextower 2, developed for liquid-lead corrosion resistance. RE = rare earths. All values are given in wt%.

Material (%wt)	Fe	Cr	Al	Si	Mn	C	Ni	RE
Nextower 1	BAL	9.5–13	3.8–4.2	<0.5	<0.7	<0.08	<0.5	0.33
Nextower 2	BAL	11–14	3.2–4.2	1–2	<0.7	<0.08	<0.5	0.59

**Table 2 materials-14-03261-t002:** Preliminary structural alloy list for the SOLEAD system, (% wt).

Material	C	Si	Cr	Ni	Al	Ti	W	Co	Cu	Nb	Mo	Fe
253MA	0.08	1.6	21	11	-	-	-	-	-	-	-	BAL
Sanicro 25	<0.1	0.2	22.5	25	-	-	3.6	1.5	3.0	0.5	-	BAL
Sanicro 31HT	0.07	0.6	20.5	30.5	0.5	0.5	-	-	-	-	-	BAL
Steel 800H	<0.1	<0.6	21	31	<0.6	<0.6			0.5			BAL
IN617	0.1	<1.0	22	44.5	1.3	-	-	10–15	-	-	9.0	<3.0

**Table 3 materials-14-03261-t003:** Creep property comparison for some structural steels, (σ: creep-stress at temperature T and at given time).

Material	Creep (10,000 h)	Creep (100,000 h)	Ref.
253MA	T = 800 °C—σ = 28 MPa;	σ = 15 MPa	Sandvik 253MA data sheet [35]
Sanicro 25	T = 800 °C—σ = 50 MPa	σ = 25 MPa	Sandvik Sanicro 25 data sheet [36]
Sanicro 31HT	T = 800 °C	σ = 27 MPa	Sandvik Sanicro 31HT data sheet [37]
Steel 800H	T = 815 °C—σ = 36 MPa	σ = 27 MPa	Specification Sheet: Alloy 800H/800HT, (UNS N08810, UNS N08811) [38]
IN617	T = 815 °C—σ = 69MPa; T = 870 °C—σ = 40 MPa	-	IN617—Special Metals Data Sheet, [39]

**Table 4 materials-14-03261-t004:** Main deposition parameters.

Technique	Coating Speed (mm/s)	Diameter of Wire (mm)	Number of Passes (Nr)	Mean Total Thickness (mm)
Gas Tungsten Arc Welding	20	1.3	4	5.1

## Data Availability

All the data is available within the manuscript.

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
