# Peer review of "Materials for High Temperature Liquid Lead Storage for Concentrated Solar Power (CSP) Air Tower Systems"

_materials, 2021, doi:10.3390/ma14123261_

Round 1

Reviewer 1 Report

This is an interesting study that fits within the scope of Materials - here a few suggestions to further improve this work:

-Don"t use abbreviation in title only, e.g. Concentrated Solar Power (CSP)...
-make sure the affiliations - etc. are formatted correctly
-the 560C are the limit of "solar salt" - I think this can quickly be mentioned since other salts allow for larger Tmax
-No need to mention the funding in the abstract - it belongs at the end of the paper and is rather confusing here...
-last sentence(s) of your abstract can provide a little outlook on the results of this study - not just what you did but if you managed to mitigate corrosion and how well that works...

-Choice of keywords seems a bit odd - why Khantal? - please check...

Introduction:
-again no need for the project number - if you want to refer to the project fine - don't make it too technical though
-again - solar salt limits your temperature to 565C - other salts do not. They dicuss using different salts and other liquids (https://doi.org/10.3390/pr5040067)
- I understand it is just a short communication but you might want to provide a bit of background - 1 to 2 sentences why increasing the temperature is actually relevant, see (https://doi.org/10.1016/j.energy.2020.117373)
- I find the project idea very interesting that you want to use knowledge from nuclear plants to improve these heat exchangers - how did the nuclear people solve it? or is it not solved yet?
- Fig. 1 use higher quality image - how does the heat exchanger exactly work - is this a tube-shell design? what is the pressure? "heat to user?" in what form? steam? please explain this a bit better even if your focus is on the corrosion experiments...

So the results are good, but this is really unstructured and messy at the moment.

Where is chapter 4?
Where is your methods chapter?

Why do you include 19 new references in the conclusions??? where you should not present new data/analysis?
If this is really relevant put it in the introduction

Restructure the results, etc. after Introduction - it is again really messy at the moment, but the data is good and deserves to be published.

Hope that helps

Author Response

Dear Reviewer,

Thank you for your very appreciated and agreed comments. We have taken care about your suggestions and please find the answers in the red text that has been added in the reviewed submission.

Your constructive comments have helped and represent the expectation for improving the paper quality from a reviewer,

Thank you again

Reviewer 2 Report

This manuscript show the study on the materials for high temperature liquid lead storage for CSP air tower systems. However this manuscript is not acceptable to publish in Materials due to the following. 

1-Structure of manuscript has not been arranged in an acceptable format. It has lack of clear section for materials and method (methodology).

2-There is no sufficient and strong discussion highlighting the finding which could be compared with similar studies.

Author Response

Dear Reviewer,

We have taken care about your genral suggestions and please find a deeply reviewed text.

This communication, is focused to highlight the experimental results that show for the first time a result of liquid lead corrosion resistance alloy at temperature of 850°C/1000h for a pilot plant. The differences with previous literature have been now presented in a clearer way (we hope) and we are convinced of the innovative content of the results.

In any case, thank you for the your help.

Reviewer 3 Report

The article was written very hurriedly and quite carelessly, it requires more data to contribute, in order to not be a report on the activities carried out. All my comments are shown in the submitted PDF file

Author Response

Dear Reviewer,

we have answered to your notes on the pdf paper text. You can find a deeply revised text that has answered to your helpful comments.

Thank you again for your help.

Round 2

Reviewer 1 Report

Good work - all comments have been addressed. I am OK with this being published now. Please make sure you move the references from the abstract in the introduction. You cannot have references in the abstract

Author Response

Thank You for your comment. I have removed the references from the abstract. Rest Regards.

Reviewer 2 Report

Revision is acceptable

Author Response

Thank You for your help in improving the paper quality